# The Effect of Dietary Supplementation on Aggressive Behaviour in Australian Adult Male Prisoners: A Feasibility and Pilot Study for a Randomised, Double Blind Placebo Controlled Trial

**DOI:** 10.3390/nu12092617

**Published:** 2020-08-27

**Authors:** Colin H. Cortie, Mitchell K. Byrne, Carole Collier, Natalie Parletta, Donna Crawford, Pia C. Winberg, David Webster, Karen Chapman, Gayle Thomas, Jean Dally, Marijka Batterham, Anne Marie Martin, Luke Grant, Barbara J. Meyer

**Affiliations:** 1School of Medicine, Lipid Research Centre, Molecular Horizons, University of Wollongong, Wollongong, NSW 2522, Australia; colinc@uow.edu.au (C.H.C.); nutribrain@tpg.com.au (D.W.); 2Illawarra Heath and Medical Research Institute, Wollongong, NSW 2522, Australia; 3School of Psychology, University of Wollongong, Wollongong, NSW 2522, Australia; mbyrne@uow.edu.au; 4South Coast Correctional Centre, Nowra, NSW 2541; Australia; carole.collier@opstar.com.au (C.C.); donna.crawford@justice.nsw.gov.au (D.C.); Karen.Chapman@scu.edu.au (K.C.); gayle.thomas@justice.nsw.gov.au (G.T.); Jean.Dally@justice.nsw.gov.au (J.D.); 5Allied Health & Human Performance, University of South Australia, Adelaide, SA 5001, Australia; Natalie.Parletta@unisa.edu.au; 6Venus Shell Systems and Shoalhaven Marine & Freshwater Centre, University of Wollongong, Nowra, NSW 2541, Australia; pia@venusshellsystems.com.au; 7Statistical Consulting Service, University of Wollongong, Wollongong, NSW 2522, Australia; marijka@uow.edu.au; 8Corrective Services New South Wales, Sydney, NSW 2000, Australia; anne-marie.martin@justice.nsw.gov.au (A.M.M.); Luke.Grant@justice.nsw.gov.au (L.G.)

**Keywords:** n-3 LCPUFA, aggressive behaviour, omega-3 index, prisoners, diet

## Abstract

This study aimed to assess the feasibility of conducting a nutrition trial in adult male prisoners. Adult male prisoners were recruited for a 16-week randomised control trial comparing the effect of ingestion of omega-3 long chain polyunsaturated fatty acids (n-3 LCPUFA) and multivitamin supplements versus placebo on aggressive behaviour. The baseline and post-intervention assessments from the participant blood samples were the erythrocyte n-3 LCPUFA levels as well as measures of aggressive behaviour determined through institutional records of misconduct (IRM), the Inmate Behaviour Observation Scale (IBOS), and questionnaires. A total of 136 adult male prisoners consented to the study with a retention rate of 60%, and 93% of blood samples were successfully collected. The IRM and IBOS scores were collected for 100% of participants, whilst 82–97% of participants completed the questionnaires. From the baseline data, the Odds Ratio shows that prisoners are 4.3 times more likely to have an IBOS >2 if they are below the 6% cut off on the omega-3 index. Both groups improved across all outcome measures and, at the current sample size, no significant differences were seen between them. A power calculation suggests a total sample size of 600 participants is required to detect the effects of this dietary supplementation, and that this supplementation study is feasible in a Correctional Centre. Important criteria for the exclusion and consideration of logistics and compliance are presented.

## 1. Introduction

Anti-social and aggressive behaviour has been found to be associated with poor diets, particularly diets deficient in omega-3 long chain polyunsaturated fatty acids (n-3 LCPUFA): eicosapentaenoic acid (EPA) and docosahexaenoic acid (DHA) [1,2]. A recent meta-analysis found that dietary supplementation with n-3 LCPUFA led to a decrease in aggressive behaviour for a diverse range of cohorts including children, juveniles, adults, and the elderly [3]. Since this review, n-3 LCPUFA supplementation was found to decrease aggression in healthy men [4], violent patients with schizophrenia [5], and children [6,7]. These results suggest that dietary supplementation with n-3 LCPUFA may be a generalisable nutritional adjunct treatment for aggression. Such a treatment is of particular interest for use in prisons, as violence in incarcerated populations has a great economic and personal cost for prisoners, staff, and the wider community [8].

The supplementing of prisoner diets with n-3 LCPUFA was first examined by Gesch et al. in a placebo-controlled, double-blind study of 231 adult prisoners in the United Kingdom [9]. In this study, participants in the n-3 LCPUFA group were given a multivitamin and a dietary supplement containing 124 mg of n-3 LCPUFA per day [9], which is a relatively low dose [10]. This dietary supplementation led to a 30% reduction in the number of institutional records of misconduct (IRM) for violent offences compared to a control group, but no differences were found in the psychometric measures of aggression between the groups. A similar outcome was reported in a study of 221 prisoners in the Netherlands, who were given multivitamins and 800 mg of n-3 LCPUFA per day, with a 34% reduction in IRM but no differences in the psychometric measures of aggression [11]. More recently, Raine et al. reported that n-3 LCPUFA supplementation of 840 mg per day led to a decrease in aggression, particularly reactive and impulsive aggression, in a population of young male offenders [12].

While these results are promising, only a limited number of studies have investigated the impact of n-3 LCPUFA supplementation on prisoner aggression. Furthermore, previous studies did not assess the blood levels of n-3 LCPUFA at baseline or post-intervention to confirm an intervention effect, with both Gesch et al. and Zaalberg et al. identifying this measurement as an important consideration for future work [9,11]. There are several important reasons to measure the blood levels of n-3 LCPUFA. Human tissue exhibits a substantial variability in the uptake of n-3 LCPUFA across individuals [13], and reporting n-3 LCPUFA intake from the consumption of supplements is not sufficient to determine levels [13,14]. Furthermore, the International Society for the Study of Fatty Acids and Lipids (ISSFAL) recently released their Official Statement Number 6, stating the importance of measuring the blood n-3 LCPUFA levels in research [15]. In addition, participants with high baseline n-3 LCPUFA levels are unlikely to benefit from additional dietary n-3 LCPUFA [10], and hence measuring both baseline and post-intervention n-3 LCPUFA levels is vital.

One standard method used to measure the n-3 LCPUFA in blood is the Omega-3 Index, which is calculated as EPA and DHA expressed as the percent of total erythrocyte fatty acids [16]. The Omega-3 Index has been used widely in numerous studies and has been postulated as a risk factor for cardiovascular disease [16]. The Omega-3 Index is emerging as a risk factor for mental health, with low levels associated with depression and schizophrenia [17] and people at ultra-high risk of psychosis [18]. We previously reported a negative correlation between the Omega-3 Index and aggressive behaviour in Australian prisoners [19], while Miles et al. [20] reported that violent offenders released into the community and provided 3 g of n-3 LCPUFA daily had an increased Omega-3 Index and reduced post-release criminal behaviour. Thus, the evidence to date would suggest that the Omega-3 Index is a useful biometric to evaluate the efficacy of omega-3 supplementation in people with aggressive behaviour.

Institutional records of misconduct (IRM) are the most widespread measure of violent and aggressive behaviour used within prisons [21,22]. However, variability across jurisdictions suggests that IRM may not be a reliable and/or sensitive measure of aggression in prisons, despite their widespread use. The Aggression Questionnaire (AQ) [23] is another measure commonly used in prison research [21], but the correlations between the AQ and IRM are low [11]. While the use of the AQ overcomes the jurisdictional differences that limit the IRM, some researchers have argued that the AQ measures impulsivity and reactivity rather than aggressive traits, and may therefore be of limited use for persistent violent offenders [24]. To overcome these limitations, we developed the Inmate Behavioural Observation Scale (IBOS), which translates the routinely collected case notes on prisoners’ behaviour into a numerical score [25]. Furthermore, attention deficit and hyperactivity disorder (ADHD) is highly over-represented within offender populations and is correlated with impulsive behaviour and aggression [26]. ADHD has been associated adult difficulties with addictions and a propensity for criminal activity [27], and therefore it is important to measure attention deficit disorders (ADD) in addition to aggressive behaviour.

Research in a prison environment can be challenging for ethical, organisational, and logistical reasons [28]. As such, a feasibility trial is required prior to undertaking a larger multi-centre trial. The aims of the present study were to examine the feasibility of (1) the recruitment and retention of participants, including reasons for dropouts; (2) the logistics of blood collection, storage, and processing; (3) compliance with self-administered capsules and potential confounds; (4) the logistics of obtaining outcome measures, including the IBOS, IRM, AQ, and Brown’s Attention Deficit Disorder Scale (BADDS); (5) the effect of omega-supplementation on aggressive behaviour and ADD symptoms; and (6) using this information for a sample size calculation for a multi-centre study.

## 2. Methods

### 2.1. Access to Correctional Centre and Study Design

Undertaking research within correctional environments in any jurisdiction requires a collaborative relationship with a correctional services executive to ensure that the research has benefit for both prisoners and the correctional service provider. Thus, the researchers approached the assistant commissioner Correctional Services New South Wales (CSNSW) to collaborate on the feasibility of conducting a randomised controlled trial investigating the effects of n-3 LCPUFA on aggressive behaviour in adult male prisoners at the South Coast Correctional Centre (SCCC) in Nowra, NSW. Funding for the trial was achieved by a University of Wollongong Partnership Grant with CSNSW as the partner. The supplements were supplied as in-kind support by NuMega Ingredients (placebo and fish oil capsules) and Nutrition Care Pharmaceuticals Pty Ltd. (the multivitamin supplements). A Correctional Officer staff member fulfilled the role as Project Officer over and above her usual duties with the support from fellow officers, the governor of SCCC, and the assistant commissioner CSNSW. This study is reported using the CONSORT Pilot Study And Feasibility Extension Statement [29] and the associated checklist.

The study was a parallel randomised control trial with an allocation of 1:1 between a group receiving either a multivitamin and the placebo or multivitamin and the n-3 LCPUFA capsules. The participants were randomly assigned to either the treatment group receiving both n-3 LCPUFA and multivitamins or a control group receiving placebo oil and the multivitamins, with both groups receiving active multivitamins because an identical size/shape/colour placebo for the multivitamins was not available. The intervention period was 16 weeks. Blood collection and data collection occurred at baseline and the completion of the study. The participants were remunerated with $10 after all the baseline data collection was achieved and $10 after the completion of data collection at the end of the intervention, and the payment went into their resettlement account.

This study was approved by the Department of Corrective Services NSW, Australia Ethics committee (11/93185), and the University of Wollongong Human Research Ethics Committee (NSA13/004). The study was registered at the Australian New Zealand Clinical Trials Registry (ACTRN12613000734763).

### 2.2. Recruitment of Study Participants

Eligible adult (over 18 years) male participants were recruited from the SCCC, Nowra, New South Wales (NSW), Australia, between 27 June 2013 and 25 November 2013. The SCCC has both maximum and minimum-security prisoners. Researchers, together with staff from SCCC, approached prisoners and explained the research project. A total of 136 study participants provided written consent to participate in the trial and were recruited in one day. The success of this recruitment was supported by the participants’ understanding of personal benefits: a perception of increased muscle strength [30] and receiving compensation ($10 at baseline and $10 at the end of the trial) for inconvenience. Those participants randomised to the placebo group were offered a jar of fish oil capsules upon completion of the study. See the consort diagram (Figure 1) for recruitment, allocation, assessments, and follow up. There was no maximum number imposed on recruitment.

Upon consenting, the enrolled participants were allocated a unique study ID. These IDs were block randomised to X or Y groups using the RALLOC command in STATA v 11.0, and were therefore allocated by chance. The determination of study conditions was conducted by a staff member independent of the research group who allocated group X to the placebo group and group Y to the omega-3 group. The study was unblinded after data analysis was completed.

### 2.3. Blood Collection and Analysis

Non-fasted blood samples were collected in EDTA tubes by nursing staff from a private pathology service in the clinical area of the South Coast Correctional Centre at baseline and after 16 weeks of intervention. An inventory of blood collection consumables, including the number of needles, was provided to SCCC security prior to the blood collection days. On the days of blood collection, all the consumables were counted going into and out from the Correctional Centre. The SCCC staff brought prisoners from various sectors of the Correctional Centre to the location of the blood collections. When that was not possible, the blood collection team moved to different sections of the prison in order to take the blood samples from the enrolled prisoners. The blood was stored on ice and transported to the University of Wollongong the same day of blood collection.

Plasma was separated from packed erythrocytes by centrifugation for 10 min at 2000× *g* at 4 °C on the same day of blood collection. Aliquots of plasma and erythrocytes were frozen at −80 °C. The erythrocyte samples were prepared for fatty acid analysis, as previously described [31,32]. A total of 1 μL of each sample was analysed by flame-ionisation gas chromatography (model GC-17A, Shimadzu, Kyoto, Japan) on a 50 m × 0.25 mm internal diameter capillary column. Individual fatty acids were quantified using the Shimadzu analysis software (Class-VP 7.2.1 SP1, Kyoto, Japan) and identified by comparison with a fatty acid standard mixture (Nu-Chek Prep, Waterville, MN, USA; Sigma Aldrich, Castle Hill, NSW, Australia). The Omega-3 Index was calculated as the sum of eicosapentaenoic acid (20:5n-3) and docosahexaenoic acid (22:6n-3) and expressed as the mol percent of total erythrocyte fatty acids [16].

### 2.4. Dietary Supplements, Compliance, and Potential Confounders

The omega-3 and placebo oil capsules were provided by the NuMega Ingredients Pty Ltd. The omega-3 supplements were fish oil containing 32% EPA (20:5n-3) and DHA (22:6n-3), and the placebo was sunola oil which did not contain EPA or DHA (Table 1). The omega-3 and placebo capsules were identical in appearance and size. Both the placebo and omega-3 group were provided with a dose of 1 g of oil per day in three capsules, equivalent to a total dose of omega-3 fatty acids of 960 mg/day and a multivitamin supplied by Nutrition Care Pharmaceuticals (Table 2). A SCCC officer provided participants with blister packs containing 7 days of supplements each week. This was repeated at the start of each week for the 16 weeks of the study intervention. Compliance was assessed by the returned blister packs. At the end of the 16-week intervention, the researchers asked participants if they thought they were on fish oil or placebo. Potential confounders for this study include the purchasing of foods/supplements containing high levels of n-3 LCPUFA, like seafood rich in omega-3 fatty acids e.g., sardines and salmon, and therefore prisoner discretionary purchase of cans of tuna, sardines, salmon and fish oil supplements was limited.

### 2.5. Outcome Measures

The primary outcome measure, the IBOS, was scored by a Correctional Services Officer using case notes from the Offender Integrated Management System, an electronic database used by Correctional Centres across the state of NSW to record information about prisoners. Case notes were scored as previously described [25]. The IBOS was scored for a baseline period of four weeks prior to the intervention, and final data were collected in the four weeks at the end of the intervention. During this time, the IRM were also collected by a Correctional Services Officer using case notes from the Offender Integrated Management System. Improvement in these measures was defined as any decrease between baseline and post-intervention.

The psychometric measures used were the self-report AQ [23] and the BADDS [33] at baseline and post-intervention. Improvement was defined as a 5-point reduction in the AQ Total or BADDS Total between the baseline and final measures. The questionnaires were administered in groups with the aid of SCCC staff psychologists, or the prisoners completed the questionnaires in their cells and returned the completed questionnaires the next day to the supporting SCCC officer.

### 2.6. Statistics

To compare group differences at baseline, Student t-tests were used for age; a Mann–Whitney U test was used for the baseline Omega-3 Index due to the non-normal distribution of this data; and Chi-Square tests were used for the ethnicity, education, and identification of participants as aggressive. Chi-square tests for association were conducted to compare the proportion of participants who saw some improvement in their measures of aggression between study groups. The odds ratio (OR) was calculated according to Bland and Altman [34].

## 3. Results

### 3.1. Recruitment and Retention

A total of 136 participants were recruited and consented to the study, and 131 participants were randomised (Figure 1). There were no differences in the baseline characteristics of the study groups (Table 3). Only 6.2% of the placebo group and 13.4% of the omega-3 group were considered aggressive at baseline, as measured by IRM. In comparison, 42.2% of the placebo group and 34.3% of the omega-3 group were considered as aggressive by the IBOS score, and 31.2% of the placebo group and 40.3% of the omega-3 group were considered aggressive, with an AQ Total score of 60 or higher. The likelihood of participants having ADD symptoms at baseline, as measured by the BADDS Total score of 55 or higher, was 26.6% for the placebo group and 29.9% for the omega-3 group.

The retention rate of participants was 61% for the placebo group and 58% for the omega-3 group, with no significant differences between the groups (*p* = 0.87). The primary cause for attrition for both groups was involuntary withdrawal due to release or transfer to other Correctional Centres, but three participants in the omega-3 group were excluded for non-compliance after reports by correctional staff that the participants were hoarding capsules (Figure 1). As this hoarding behaviour only occurred in the omega-3 group, it is likely that the blinding of the omega-3 and placebo capsules was imperfect. No adverse events were reported during the study.

### 3.2. Blood Collection and the Omega-3 Index

Following appropriate security protocols set out by CSNSW, it was feasible for an external provider to visit a Correctional Centre to take the necessary blood samples at baseline and post-intervention. No prisoners refused to provide a blood sample at baseline, but five prisoners out of 136 (4%) did not give blood at baseline because they were either in court or were moved to another Correctional Centre. One prisoner from 131 prisoners (1%) refused to provide a blood sample after the intervention, and 8 (6%) were not available for blood collection at the end of the intervention. Participants from both groups had similar distributions of the Omega-3 Index at baseline (Figure 2A), but by the end of the study the distribution of the omega-3 group had shifted higher up the Omega-3 Index, while that of the placebo group had not (Figure 2B). Of the 75 participants that completed the study, the median (IQR) Omega-3 Index (expressed as mol % of total fatty acids) at baseline and post-intervention in the placebo group (n = 39) was 5.1 (4.2, 7.2) and 5.5 (4.2, 6.6) with a mean change of 0.4 which was not significant (*p* = 0.80). The median (IQR) omega-3 index at baseline and post-intervention in the omega-3 group (n = 36) was 4.6 (4.3, 5.7) and 7.8 (6.6, 8.7) with a mean change of 3.2 which was significantly higher than baseline (*p* < 0.0001).

### 3.3. Compliance to Treatment and the Omega-3 Index

Determining compliance to treatment as assessed by the return of blister packs was unsuccessful. The use of blister packs was unreliable due to many not being returned by participants, and a lack of supervision and monitoring of compliance by staff.

At baseline, 25% of the placebo group and 17% of the omega-3 group had an Omega-3 Index of 8% or higher (Figure 2A). By the end of the study, 20% of the placebo group and 59% of the omega-3 group had an Omega-3 Index of 8% or higher (Figure 2B). The change in the Omega-3 Index was calculated for participants who completed the study, with no change for most participants in the placebo group and a change of 2% or more for most participants in the omega-3 group (Figure 2C). A total of 16% of participants in the placebo group had an increase of 2% or more in their Omega-3 Index, while 11% of participants in the omega-3 group did not have an increase. No adverse effects were reported for either group. At the end of the trial, those in the placebo group guessed 50:50% correct or incorrect, whilst the fish oil group guessed 60:40% correct or incorrect. Comments made by the study participants include “I was on placebo because I couldn’t smell anything when I cut it open”, “on active because I could taste it”, and “on active because I could smell it”. Other anecdotal comments included “better training”, “shoulder pain improved”, “better sleep”, and “better joints”.

### 3.4. Collection of Outcome Measures

The IBOS scores and IRM were successfully collected for all the participants at baseline and at the completion of the study. Data collection was lower for the psychometric measures, with the AQ collected for 89% of the control group and 88% of the omega-3 group at baseline, while post-intervention was 98% and 94% completion for the control and omega-3 groups, respectively. Similarly, the BADDS was collected for 88% and 82% of the control and omega-3 groups at baseline, and 97% and 92% post-intervention. The incomplete collection of questionnaires was due to participants being unavailable during the collection periods. It was also reported anecdotally that some participants found completing the questionnaires difficult due to poor literacy levels. It was estimated that 15–20% of the prisoners needed help with the completion of the questionnaires.

### 3.5. Changes in Aggression

The placebo and omega-3 groups had similar levels of improvement across all measures, with the highest percentage of improvement seen in the IBOS and the lowest seen in IRM (Figure 3A). No significant differences were seen between groups for the improvement of aggression, as measured by the IRM (χ^2^(1) = 0.14, *p* = 0.905), IBOS scores (χ^2^(1) = 0.830, *p* = 0.362), AQ Total (χ^2^(1) = 0.026, *p* = 0.817), and BADDS Total (χ^2^(1) = 0.060, *p* = 0.635). The majority of participants in this study were non-aggressive at baseline (Table 1), and thus no improvement in aggression was possible.

The subset of participants characterised as aggressive by their baseline IBOS were analysed for the omega-3 group (n = 11) and the placebo group (n = 18) (Figure 3B). In this subset, there was a trend toward greater improvement in aggression for the omega-3 group than the placebo group for all measures. The difference in improvement between groups was 7% for the IBOS, 25% for records of misconduct, 19% for the AQ Total, and 2% for the BADDS Total. Due to the small sample number in this subset, however, no significant differences were reported for the IRM (χ^2^(1) = 2.653, *p* = 0.103), IBOS (χ^2^(1) = 0.333, *p* = 0.566), AQ Total (χ^2^(1) = 0.800, *p* = 0.371), or BADDS Total (χ^2^(1) = 0.24, *p* = 0.910).

The OR shows that prisoners are 4.5 (95% CI 1.0, 20) times more likely to have an IBOS of >2 if they are below the cut off of 6% (Omega-3 Index). Prisoners are (OR) 2.0 (95% CI 0.65, 5.8) times more likely to have IBOS scores indicating prosocial behaviour if they are above the cut off of 6% (Omega-3 Index).

### 3.6. Power Calculation

A power calculation for a future multi-centre trial was conducted, taking into account the need for different study sites. Given the nature of prison populations, it is likely that there will be an effect of prison location in the analysis, and therefore a design effect accounting for the intra-cluster correlation (ICC) is incorporated in the sample size estimation, with a conservatively estimated ICC of 0.03. Based on the power calculation, a total sample size of 550 enrolled participants and a retention of 360 participants would allow a difference of 25% IBOS scores to be detected with 80% power and an alpha level of 0.05.

## 4. Discussion

This study examined the feasibility of a randomised, placebo-controlled dietary intervention with omega-3 capsules providing a total dose of 960mg EPA + DHA/day and a multivitamin to improve aggressive behaviour in prisoners at an Australian Correctional Centre. An important consideration for a study of this type is the need for institutional support from the executive team of the prison system. In this study, support was provided by the Deputy Commissioner of CSNSW. Due to the complexity of prisons, support from frontline staff is also required to ensure access to participants and their data. This support was particularly important for blood collection, which required both institutional authority and site-specific knowledge. Based on this study’s findings, future trials will require a dedicated Correction Services Officer at each project site who has both the imprimatur to access all areas of the prison and the time to dedicate entirely to the research.

During recruitment, many participants who were approached consented to the study due to incentives such as reimbursement, the break from routine, and anticipated benefit to muscle strength from the n-3 LCPUFA supplements provided. For the placebo group, n-3 LCPUFA supplements were provided following the study as an incentive in case they felt they missed out upon it being revealed that they were on the placebo. Only about a third of participants were considered aggressive at the baseline. A lower percent of participants were found to likely have a diagnosis of ADD using the BADDS questionnaire at baseline. The low levels of baseline aggression and ADD in the study cohort resulted in little change in these characteristics over the intervention. The retention rate in this study, approximately 60%, was lower than the 75% retention reported in the United Kingdom [9] and 68% in the Netherlands [11]. Attrition was primarily due to the participants being forced to withdraw by transfers to other correctional centres or parole. Additional criteria for a future study would be the selection of participants likely to remain in the centre for the duration of the study, and the recruitment of participants assessed as aggressive at baseline.

Despite the added logistical complexity of collecting blood, the blood sampling from an external pathology provider was successful, although several participants were lost at baseline or post-intervention due to being unavailable for blood collection. The analysis of this blood at baseline identified participants with an Omega Index of 6% or higher, and these participants were unlikely to benefit from dietary n-3 LCPUFA supplementation due to a potential ceiling effect [10,16]. The collection of blood at the end of the intervention allowed a physiological change in the Omega-3 Index to be determined for each participant [14,15]. Based on dosage studies, a minimum change of 2% in the Omega-3 Index would be expected for the n-3 LCPUFA group, and no change would be expected for the placebo group [10,35]. This physiological assessment of compliance was important due to several confounding factors identified for compliance. Given the potential of ceiling effects in terms of improving the Omega-3 Index, it was determined that a baseline index of less than 6 should be included in participant selection for a multi-centre trial.

The return of used blister packs, which was the original measure of compliance proposed, was found to be unreliable with many participants not handing their blister packs back. In some cases, this was due to capsules being traded or hoarded as a commodity. The group receiving the n-3 LCPUFA capsules had a better than equal chance of guessing which treatment they were on due to the smell of the oil, with several participants in this group excluded for the trading or hoarding of capsules. Another cofounder was that participants could augment their normal diet with tuna or other foods high in n-3 LCPUFA through purchases or buy-ups from the centre. Future studies would benefit from a greater degree of staff oversight of capsule compliance, including supplement delivery directly to participants, watching participants take the supplements, and improving the blinding of the treatment by flavouring both placebo and n-3 LCPUFA supplements to mask the “fish smell”. Future studies would also benefit from restricting the purchase of n-3 LCPUFA-rich foods/supplements from buy-up items at the centres.

Data collection for the IBOS and IRM was done for all the participants at both baseline and post-intervention, as the Correctional Officer had access to the Offender Integrated Management System (OIMS). However, the AQ and BADDS questionnaires were collected for about 90% of the participants. Self-report questionnaires such as the AQ and BADDS require participants’ cooperation and time, which can be a limiting factor for participants who are unwilling or unable to complete the questionnaires. There were also several anecdotal reports of participants struggling to fill out their AQ and BADDS due to limited literacy skills. This may lead to inaccurate responses for these measures. Differences were also seen between the sensitivity of the measures, with more participants classified more aggressive according to the IBOS and AQ than IRM. This may be explained by the relative low incidence rate of IRM [25]. The use of multiple measures of aggression was therefore shown to be feasible, with the IBOS more sensitive than IRM and more reliably collected than self-report questionnaires. A future multicentre trial should ensure that all the participants are supported to complete the AQ and BADDS.

There was no treatment effect between the two groups across the intervention. Previous omega-3 interventions in prisoners reported no significant decrease in the AQ scores, but did report a decrease in IRM for omega-3 groups [9,11]. One possible reason for a similar decrease in IRM not being found was that this feasibility was not statistically powered to find a significant result between groups, and that most participants were not aggressive at baseline and therefore were not able to improve on this behaviour. The selection of aggressive prisoners for intervention studies is difficult, with past interventions selecting their cohorts by criminal convictions, self-reports, or IRM [22]. In this study, the IBOS was used to identify the subset of aggressive participants for analysis, which was approximately 25% of the total cohort. Ideally, this subset would only include participants who complied with treatment, as measured by their Omega-3 Index, but due to small samples sizes in this subset all the participants were examined by their intention to treat. Improvements were seen for both groups for measures of aggression, which may be attributable to the multivitamins given to both groups. The largest improvement for both groups was seen for the IBOS, although the reason for this remains unclear. Overall, a greater trend in improvement in aggression was seen for the n-3 LCPUFA group than the placebo group, although no significant differences were observed. A multicentre study should ensure that only participants with a baseline measure of aggressive behaviour are included in the study.

Several limitations were identified in this study design which will need to be addressed in future studies. The IBOS has not been subject to peer review. Both the placebo and n-3 LCPUFA groups were given a multivitamin due to no acceptable multivitamin placebo being available at the time of the study, and the need to commence the study to meet the timeline milestones. Whilst recognising that this study design is not ideal, the main aims of this study was to assess the feasibility of recruitment/retention of study participants, logistics of blood collections and obtaining the outcome measures as well as assessing compliance. There is some evidence that dietary supplementation with n-3 LCPUFA leads to a decrease in aggression compared to supplementation with multivitamins or a placebo [36], but the provision of the multivitamin to both groups may in part explain the decrease in the IBOS scores of aggression for both groups. This study was also limited by a lack of inclusion or exclusion criteria during recruitment. For example, we did not exclude study participants that did not demonstrate aggressive behaviour and hence had a small sample size of study participants that actually had aggressive behaviour. Future studies should either exclude multivitamins or ensure an appropriate placebo and recruit prisoners with both high levels of aggressive behaviour (e.g., IBOS score of 1–5) and low levels of Omega-3 Index (e.g., less than 6%).

In summary, we have shown that this research is feasible in a Correctional Centre. In order to conduct further research in this area, the following recommendations are encouraged to be followed: (1) the need for support from and collaboration with Correctional Services; (2) the need to provide incentives to prisoners to aid in recruitment; (3) blood sampling from an external pathology provider was successful; (4) the need for a designated Correctional Officer as a Project Officer to ensure compliance; (5) the need to reduce potential confounders—e.g., restrict buy-ups of cans of tuna and n-3 LCPUFA supplements; (6) the need to train the Correctional Officers on IBOS scoring and how to complete the questionnaires; and (7) the need to have inclusion and exclusion criteria to ensure that the participants are likely to remain in the study for the study duration, are aggressive at baseline (IBOS of 1 or greater), and have an Omega-3 Index below 6%. Based on a conservative estimate, a total sample size of 550–600 enrolled participants and a retention of 360 participants would allow a difference of 25% IBOS scores to be detected with 80% power and an alpha level of 0.05. As most Correctional Centres in Australia are not able to provide such a large population, a trial of this size would require several sites.

## 5. Conclusions

This feasibility study shows that a double-blind, randomised control trial with n-3 LCPUFA supplements or a placebo is feasible in an Australian correctional centre with blood collection from participants. To achieve definitive outcomes, a designated correctional officer on secondment as a project officer per site is vital; a multicentre trial is warranted for adequate statistical power; the study design needs to have strict inclusion/exclusion criteria to maximise a significant outcome and not reach a potential ceiling effect; and blood should be taken to measure the n-3 LCPUFA levels and, hence, to verify the biological plausibility.

## Figures and Tables

**Figure 1 nutrients-12-02617-f001:**
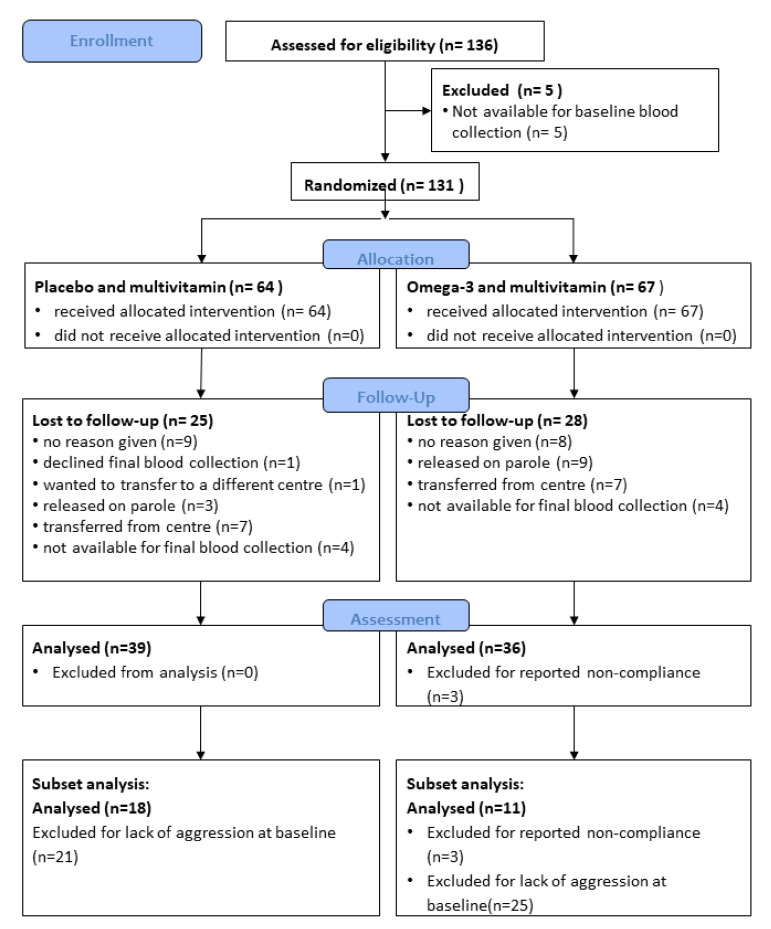
CONSORT flow diagram of recruitment, follow-up, assessment of the whole cohort, and assessment of the subset of aggressive participants, as defined by an Inmate Behaviour Observation Scale (IBOS) of 1 or greater at baseline.

**Figure 2 nutrients-12-02617-f002:**
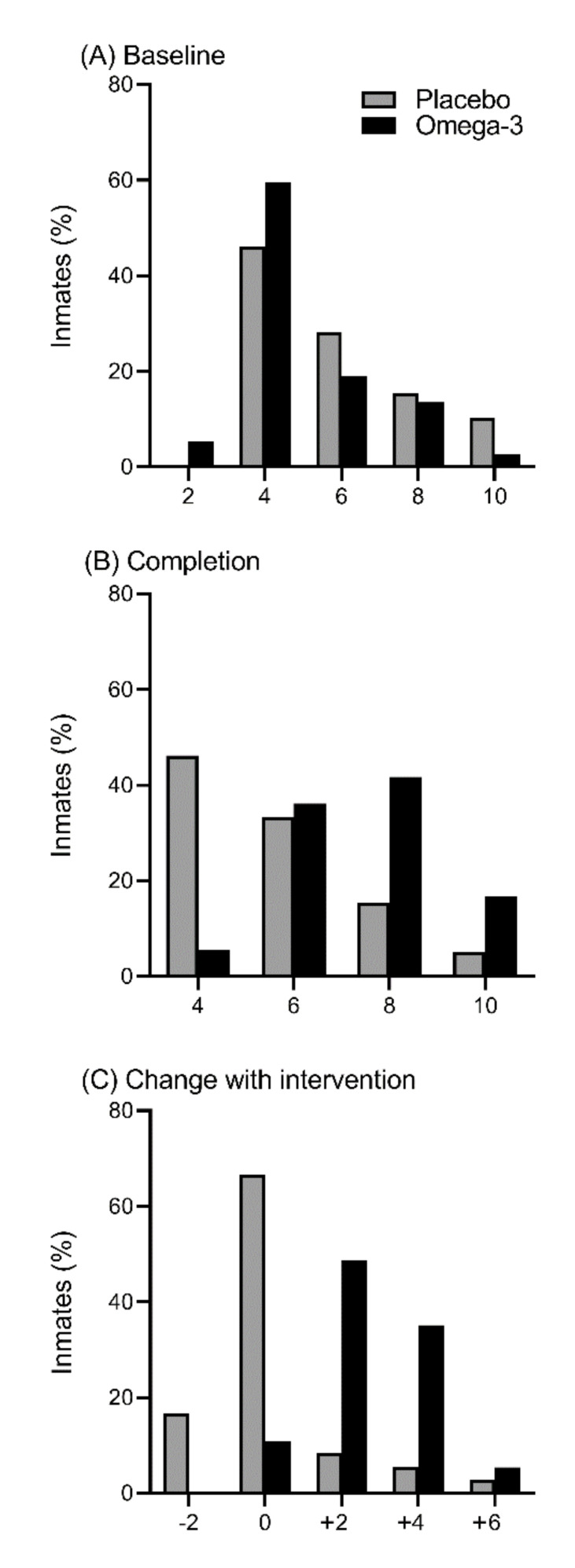
Distribution of the Omega-3 Index (**A**) at baseline (n = 131), (**B**) at the completion of the study (n = 75), and (**C**) as a change between these two time points for participants who completed the intervention (n = 75).

**Figure 3 nutrients-12-02617-f003:**
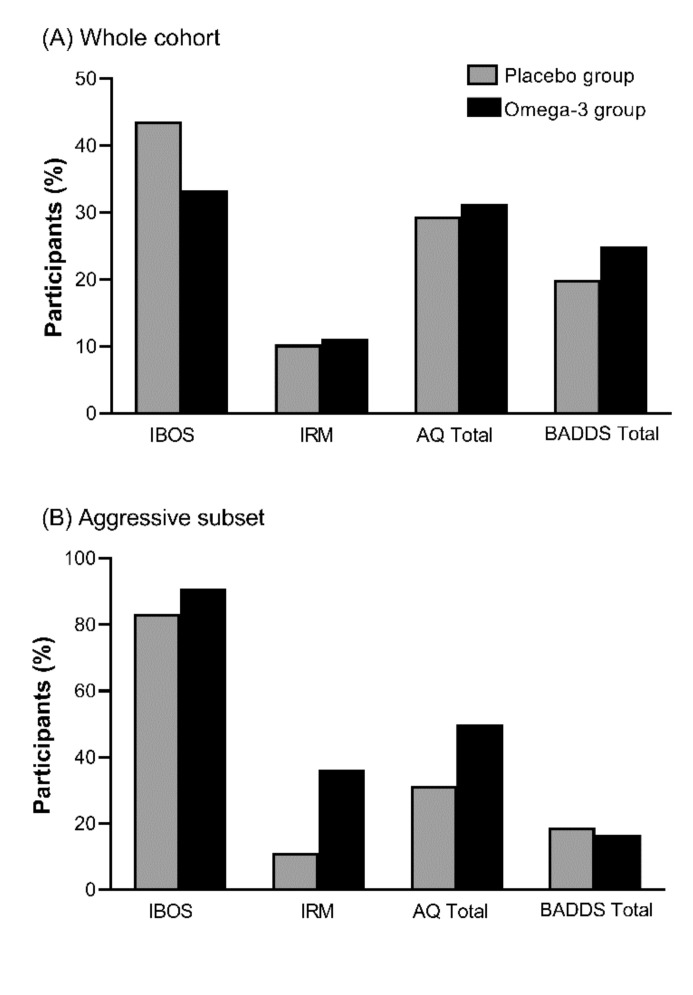
Percentage of participants who improved in aggression in (**A**) the whole cohort (n = 75) and (**B**) the subset of participants considered aggressive at baseline, as defined by a score of 1 or greater on the Inmate Behaviour Observation Scale (IBOS) (n = 29). Improvement in behaviour was measured as a decrease in the IBOS, a decrease in the number of institutional records of misconduct (IRM) over four weeks at the end of the trial, a decrease of at least five points in the Aggression Questionnaire’s Total scale (AQ Total), and a decrease of at least five points in the Brown Attention Deficit Disorder Total scale (BADDS Total). No significant differences were observed for the whole cohort or the aggressive subset.

**Table 1 nutrients-12-02617-t001:** Composition of fatty acids (mol% of total fatty acids *) present in the placebo and omega-3 capsules.

Fatty Acid	Placebo mol%	Omega-3 mol%
SFA		
14:0 myristic acid	0.1	3.0
16:0 palmitic acid	5.7	20.4
18:0 stearic acid	2.7	5.9
MUFA		
16:1 palmitoleic acid	0.1	5.4
18:1n-9 oleic acid	81.5	13.0
18:1n-7 vaccenic acid	0.0	2.3
PUFA n-6		
18:2n-6 linoleic acid	7.7	1.3
20:4n-6 arachidonic acid	0.0	1.8
22:5n-6 n-6 docosapentaenoic acid	0.0	2.0
PUFA n-3		
18:3n-3 alpha-linolenic acid	0.3	0.5
20:5n-3 eicosapentaenoic acid	0.0	5.1
22:5n-3 n-3 docosapentaenoic acid	0.0	1.0
22:6n-3 docosahexaenoic acid	0.0	26.9
Other fatty acids	1.6	2.7
∑ SFA	9.5	32.3
∑ MUFA	82.0	23.1
∑ PUFA	8.0	40.5
∑ n-6 PUFA	7.7	6.0
∑ n-3 PUFA	0.3	34.5

* Only fatty acids with an abundance of 0.1% or greater are included. SFA, saturated fatty acids; MUFA, monounsaturated fatty acids; PUFA, polyunsaturated fatty acids.

**Table 2 nutrients-12-02617-t002:** Multivitamin ingredients.

Ingredient	Amount
Vitamins	
Vitamin A	750 IU
Vitamin B1	50 mg
Vitamin B2	20 mg
Vitamin B3	10 mg
Vitamin B3	200 mg
Vitamin B5	100 mg
Vitamin B6	50 mg
Vitamin B7/Vitamin H	20 mcg
Vitamin B9	150 mcg
Vitamin B12	100 mcg
Vitamin C	50 mg
Vitamin D3	100 IU
Vitamin E	25 IU
Minerals	
Calcium	10 mg
Cobalt	25 mcg
Copper	6 mcg
Lithium	140 mcg
Magnesium	7.6 mg
Manganese	93 mcg
Potassium	2 mg
Potassium	13.6 mg
Zinc (as gluconate)	1.3 mg
Zinc (as sulphate)	7.6 mg
Other	
Amino-benzoic acid	20 mg
Betacarotene	3 mg
Betaine hydrochloride	10 mg
Choline bi-tartrate	50 mg
Inositol	25 mg
L-Glutamine	50 mg
Lysine hydrochloride	10 mg
*Scutellaria lateriflora* (Equiv. dry herb)	100 mg
*Valeriana officinalis* (Equiv. dry root)	100 mg

**Table 3 nutrients-12-02617-t003:** Participant characteristics at baseline.

	Placebo Group (n = 64)	Omega-3 Group (n = 67)	
Age			
Mean ± SD	33.3 ± 10.3	33.7 ± 12.6	*p* = 0.852
Range	18–70	19–80	
Ethnicity–number (%)			
African	1 (1.6)	0 (0)	X^2^ = 0.888
Arabic	6 (9.4)	5 (7.5)	
Asian	6 (9.4)	6 (9.0)	
Australian Aboriginal	7 (11)	7 (10)	
Caucasian	36 (56)	38 (57)	
Hispanic	3 (4.7)	2 (3.0)	
Polynesian	5 (7.8)	8 (12)	
Not stated	0 (0)	1 (1.5)	
Education–number (%)			
Primary school	3 (4.7)	3 (4.5)	X^2^ = 0.508
Lower high school	32 (50)	36 (54)	
Upper high school	14 (22)	9 (13)	
Tertiary	4 (6.2)	4 (6.0)	
Unknown	11 (17)	15 (22)	
Omega-3 Index			*p* = 0.067
Median (IQR)	4.91 (4.25, 6.68)	4.59 (4.08, 5.55)	
Range	2.44–10.02	2.27–10.3	
Identified as aggressive by measure–number (%)			
IRM	4 (6.2)	9 (13)	X^2^ = 0.169
IBOS	27 (42)	23 (34)	X^2^ = 0.355
AQ Total	20 (31)	27 (40)	X^2^ = 0.280
Likely to have ADD–number (%)			
BADDS	17 (27)	20 (30)	X^2^ = 0.856

IQR, Interquartile range (0.25, 0.75); IRM, Institutional Reprimand Misconduct; IBOS, Inmate Behavioural Observation Scale; AQ, Aggression Questionnaire; ADD, Attention Deficit Disorder; BADDS, Brown’s Attention Deficit Disorder Scale.

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
