# Peer review of "The Effect of Dietary Supplementation on Aggressive Behaviour in Australian Adult Male Prisoners: A Feasibility and Pilot Study for a Randomised, Double Blind Placebo Controlled Trial"

_nutrients, 2020, doi:10.3390/nu12092617_

Round 1
Reviewer 1 Report
This was a really well-written clear paper, and I'm grateful for the opportunity to review it. My comments are incredibly minor - I'm happy to review responses / further iterations of the paper.
My substantive comment relates to methodology. Usually, in a feasibility trial/study, a qualitative component of work would be undertaken. Can the authors comment on whether they did so. If they did, it may be worthwhile flagging this in the paper and that findings are forthcoming? If not, could the authors explain why not, and how participant reflections on the trial were captured?
Other small comments:
- How did participants receive their incentive?
- Could the authors proof read? There are a few missing words here and there.
Author Response
Reviewer 1
This was a really well-written clear paper, and I'm grateful for the opportunity to review it. My comments are incredibly minor - I'm happy to review responses / further iterations of the paper.
My substantive comment relates to methodology. Usually, in a feasibility trial/study, a qualitative component of work would be undertaken. Can the authors comment on whether they did so. If they did, it may be worthwhile flagging this in the paper and that findings are forthcoming? If not, could the authors explain why not, and how participant reflections on the trial were captured?
Other small comments:
- How did participants receive their incentive?
- Could the authors proof read? There are a few missing words here and there.
Due to the nature of the study environment, that is, restrictions in access of research personnel due to security requirements, the opportunity to acquire qualitative feedback was limited. We acknowledge that qualitative data would be useful had the research environment enabled it. However, we note that at the end of the trial we asked the study participants what supplements they were on (placebo or omega-3), and they offered us the quotes we included in the manuscript - hence the participant reflections were captured in an opportunistic way.
The study participants received their incentives in their resettlement account. We have added this to the manuscript (see lines 133-134).
The authors have proof read the manuscript as suggested.
Reviewer 2 Report
This study is certainly interesting. The effort in the proceedings must be also acknowledged. However, the main concern I have with this study is that all the power of the conclusions lay on an instrument that has not yet been validated (IBOS scale). I think the study is well-designed, interesting, and has great applicability potential. However, I would suggest the authors to wait until the IBOS scale is published to submit this work.
Author Response
Reviewer 2
This study is certainly interesting. The effort in the proceedings must be also acknowledged. However, the main concern I have with this study is that all the power of the conclusions lay on an instrument that has not yet been validated (IBOS scale). I think the study is well-designed, interesting, and has great applicability potential. However, I would suggest the authors to wait until the IBOS scale is published to submit this work.
While we agree with Reviewer 2 that the IBOS publication would better precede the feasibility and pilot paper, we are progressing our research and feel it appropriate to inform the field of our work before we progress beyond the pertinence of a feasibility/pilot paper. Further, the IBOS paper has been submitted and is under review, however the receiving journal has a longer processing time than Nutrients and we would not be able to resolve this within the agreed response time.
Reviewer 3 Report
Dear Editor-in-chief, Nutrients journal;
Greetings!
Hereby I am sending my comments related to the manuscript entitled "The effect of dietary supplementation on agressive behaviour in Australian adult male prisoners: a feasilibility and pilot study for a randomized, double blind and placebo controlled trial". Authors provide the findings from their pilot study and recommendations for future research projects.
With great interest, I read this paper thoroughly and found that its content phas some new information in terms of ways to elaborate a better research protocol susceptible to result in meaningful outcomes. Despite the good quality of the paper, I think there is a possibility to get it improved by considering and addressing satisfactorily the following concerns.
- Major points: none.
2. Minor points:
2.1. Abstract section:
-Line122-123: In this sentence, authors wrote that "The study was ....between a group receiving either a multivitamin and the placebo or multivitamin and the n-3 LCPUFA capsules". What is true is that they compared groups; however, in the above sentence they did not show this fact.
2.2. Introduction section:
- Line 44-46: "Since this review,....". In this sentence, authors present findings from previous studies conducted in viloents patients and children; but the use of the adverb "Since" to start this sentence does not make sense, I think. Instead, 'Prior to this review..', or 'Previously...' would be appropriate.
2.3. Methods section:
- Statistical analysis: - the use of Mann-Whitney U test (rank-sum test) for data analysis in this study is questionable. As you deal with IBOS, IRM which are continuous variables, I wonder why you used both Student t test and also M-W U-test. Student t test would be enough to compare the 2 groups for outcome variables that are continuous/quantitative.
2.4. Discussion section:
- The authors mentioned that there were several limitations in this study; however, it seems they showed only one limitation (related to the use of multivitamin in both groups).
- What about the reduced/small size of the sample regarding aggressive participants? As you commented, study findings showed no significant differences between study groups for a number of outcome variables . This fact is at least partially due to the small number of participants with aggressive behaviours at baseline. If so, it should be well discussed as a limitation for this study.
Author Response
Reviewer 3
Greetings!
Hereby I am sending my comments related to the manuscript entitled "The effect of dietary supplementation on aggressive behaviour in Australian adult male prisoners: a feasibility and pilot study for a randomized, double blind and placebo controlled trial". Authors provide the findings from their pilot study and recommendations for future research projects.
With great interest, I read this paper thoroughly and found that its content has some new information in terms of ways to elaborate a better research protocol susceptible to result in meaningful outcomes. Despite the good quality of the paper, I think there is a possibility to get it improved by considering and addressing satisfactorily the following concerns.
- Major points: none.
- Minor points:
2.1. Abstract section:
-Line122-123: In this sentence, authors wrote that "The study was ....between a group receiving either a multivitamin and the placebo or multivitamin and the n-3 LCPUFA capsules". What is true is that they compared groups; however, in the above sentence they did not show this fact.
The reviewer is correct as the previous sentence “This study aimed to assess the feasibility of conducting a nutrition trial in adult male prisoners.” does not explicitly state that two groups are compared, but the fact that it refers to a trial suggests that two groups might be compared. The next sentence then goes on to the describe the trial “Adult male inmates were recruited for a 16-week randomised control trial comparing the effect of ingestion of omega-3 long chain polyunsaturated fatty acids (n-3 LCPUFA) and multivitamin supplements versus placebo on aggressive behaviour.” Where clearly there are two groups being compared. Therefore, with respect, we suggest that the current wording adequately informs the reader of the study design. We note that this issue was not raised by reviewers 1 and 2.
2.2. Introduction section:
- Line 44-46: "Since this review,....". In this sentence, authors present findings from previous studies conducted in violent patients and children; but the use of the adverb "Since" to start this sentence does not make sense, I think. Instead, 'Prior to this review..', or 'Previously...' would be appropriate.
With respect, we disagree with the reviewer. The review discussed preceding this comment was published in 2016 and since that review; other authors have published further studies in 2018, 2018, 2019 and 2016. These four publications were published after the systematic review that was published in 2016.
2.3. Methods section:
- Statistical analysis: - the use of Mann-Whitney U test (rank-sum test) for data analysis in this study is questionable. As you deal with IBOS, IRM which are continuous variables, I wonder why you used both Student t test and also M-W U-test. Student t test would be enough to compare the 2 groups for outcome variables that are continuous/quantitative.
We thank the reviewer for raising this point and apologise for the confusion. MWU tests were only used for baseline omega-3 index due to the non-normal distribution of this data. The IBOS and IRM were assessed as categorical outcomes (aggressive or not) at baseline, but statistics are incorrectly stated as p values rather than X2-values. This has been corrected in the table – the values themselves do not change.
We have now clarified the use of statistics in Line 218-220:
To compare group differences at baseline, student t-tests were used for age, a Mann-Whitney U test was used for baseline omega-3 index due to the non-normal distribution of this data, and Chi-Square tests were used for ethnicity, education, and identification of participants as aggressive.
2.4. Discussion section:
- The authors mentioned that there were several limitations in this study; however, it seems they showed only one limitation (related to the use of multivitamin in both groups).
- What about the reduced/small size of the sample regarding aggressive participants? As you commented, study findings showed no significant differences between study groups for a number of outcome variables. This fact is at least partially due to the small number of participants with aggressive behaviours at baseline. If so, it should be well discussed as a limitation for this study.
We agree with the reviewer and we have added the small number of participants with aggressive behaviour at baseline to the limitations section in the manuscript (see lines 411-413).
Round 2
Reviewer 2 Report
I understand the author's reasons and their response. Unfortunately, I must insist on my position. And I maintain what I said before; this study is solid, interesting and of great applicability. I would gladly accept it on its current form... After the publication of the preceding IBOS paper.
Author Response
I understand the author's reasons and their response. Unfortunately, I must insist on my position. And I maintain what I said before; this study is solid, interesting and of great applicability. I would gladly accept it on its current form... After the publication of the preceding IBOS paper.
Upon the advice of the Academic Editor we have added the non-validated IBOS scale to the limitations section of the manuscript.